# Are Changes Occurring in Bacterial Taxa Community and Diversity with the Utilization of Different Substrates within SIR Measurements?

**DOI:** 10.3390/microorganisms12102034

**Published:** 2024-10-09

**Authors:** Yosef Steinberger, Tirza Doniger, Itaii Applebaum, Chen Sherman

**Affiliations:** The Mina & Everard Goodman Faculty of Life Sciences, Bar-Ilan University, Ramat Gan 5290002, Israel; tirza.doniger@gmail.com (T.D.); itaiiapplebaum@gmail.com (I.A.); chen.sherman1@gmail.com (C.S.)

**Keywords:** substrate utilization profile, soil fungal community, soil bacteria community, diversity

## Abstract

This research explores how the availability of substrates affects the regulation of soil microbial communities and the taxonomical composition of bacteria. The goal is to understand the impact of organic matter and substrate availability and quality on the diversity of soil bacteria. The study observed gradual changes in bacterial diversity in response to the addition of different substrate-induced respiration (SIR) substrates. Understanding the structure, dynamics, and functions of soil microbial communities is essential for assessing soil quality in sustainable agriculture. The preference for carbon sources among bacterial phyla is largely influenced by their life history and trophic strategies. Bacterial phyla like *Proteobacteria*, *Bacteroidetes*, and *Actinobacteria*, which thrive in nutrient-rich environments, preferentially utilize glucose. On the other hand, oligotrophic bacterial phyla such as *Acidobacteria* or *Chloroflexi*, which are found in lower numbers, have a lower ability to utilize labile C. The main difference between the two lies in their substrate utilization strategies. Understanding these distinct strategies is crucial for uncovering the bacterial functional traits involved in soil organic carbon turnover. Additionally, adding organic matter can promote the growth of copiotrophic bacteria, thus enhancing soil fertility.

## 1. Introduction

In terrestrial ecosystems, whether natural or human-made, most soil organisms exist in unpredictable environments, relying on random abiotic conditions to carry out their biological functions. The “pulse-reserve” model, proposed by [1], effectively explains how environmental stress influences organisms to shift between different metabolic stages, ranging from dormancy (such as cysts, anhydrobiosis, aestivation, etc.) to full activity. This ability to transition between states helps them avoid unfavorable conditions [2]. The maintenance of biodiversity in the soils of intensively developed agricultural lands is declining in the developing world with the quest to increase primary production globally parallel to developing a sustainable environment. The increase in fertilizers and a wide range of agrochemicals have been found to boost nutrient availability and primary production on one hand and to decrease soil biota functionality on the second hand. As a result, much more attention has been paid to the environmental role of natural soil biota, where using the present native soil biota is advantageous and more feasible. Their presence in high numbers and the trophic interaction between them in soil ecosystems contribute to health, function, services of the environment via mineralization, nutrient recycling, detoxification, and rising soil fertility [3,4,5]. Such remarkable activity of microorganisms is established based on their metabolic activity, diversity, versatility, and genetical adaptability of the below-ground community. Bacteria and fungi are known to be the main components of microbial biomass, and their relative importance is related to the organic matter content and soil texture. Anderson and Domsch [6] and Christensed and Funck-Jensen [7] in their study on seventeen types of agricultural and forest soil showed that the contribution of bacteria and fungi to microbial biomass averaged 25% (10–40%) and 75% (60–90%), respectively. Their activity and their interaction with soil organic matter play an important role in terrestrial ecosystem processes, which can be determined by measuring basal respiration/CO_2_ evolution. The CO_2_ evolution as a result of microbial activity is significantly related to substrate availability and nutrient demand of the microflora [8,9]; changes in microbial biomass are accompanied by changes in organic matter quality that will determine the initial amount of activity expressed by the respiration rate [10,11,12]. CO_2_ evolution from soil determines the combined respiration of fungi and bacteria, where the efficiencies of fungi and bacteria are different due to the different substrates’ utilization factors [13,14] (Vishnevetsky and Steinberger, 1996; Oren and Steinberger, 2008). The different ways bacteria utilize carbon can indicate soil health. By examining the bacterial community structure, scientists and land managers can evaluate how land use practices or environmental changes affect soil quality and make informed decisions to protect or restore ecosystems. According to Fierer et al. [15], the copiotroph-oligotroph dichotomy framework, which differentiates microbial life histories based on substrate utilization strategies, suggests that copiotrophic bacterial phyla, such as *Proteobacteria*, *Bacteroidetes*, and *Actinobacteria*, preferentially utilize glucose-C. In contrast, oligotrophic bacterial phyla, including *Acidobacteria* and *Chloroflexi*, exhibit a reduced capacity to utilize labile carbon sources [16,17,18].

Many different approaches have been used in order to determine the functions of a microbial community using selective enrichment culture since the early 1900s [19]. Due to the development of molecular tools and DNA sequencing, an opportunity to thoroughly understand bacteria and fungi communities and biodiversity patterns has appeared [20].

Considering the microbial community in the soil as allogenic, we hypothesize that any amendment as a substrate to the soil milieu will have a progressive effect on microbial biomass that will result in the composition of the microbial community, which will allow us to determine community taxa composition and diversity. This study aims to understand the effects of substrate availability and quality on bacterial community performance at the molecular level where the following apply: (1) substrate resources will alter microbial community composition, diversity, and taxa, and (2) food-source availability will trigger unknown (anonymous) microbial taxa.

## 2. Materials and Method

### 2.1. Study Site and Soil Collection

The study site is in the crop field of Kibbutz Be’erot Itzhak, Israel (32°02′05.1″ N 34°54′48.2″ E), 52 m above sea level, with a multiannual average rainfall of 489 mm. This area is characterized by a Mediterranean climate, where the average minimum temperature is 2.8 °C, and the average maximum temperature is 36.5 °C. Soil particle composition in the wheat field and control area are 77% and 58% sand, 17% and 29% silt, and 6% and 12% clay, respectively. The study site is a wheat agriculture site surrounded by natural non-planted area where both are characterized mainly as sandy texture and low clay [21].

Soil rhizosphere samples were collected randomly in the late spring, from the upper 0–10 cm layer during the early hours (*n* = 3), using a soil core (5 cm diameter) from a 300 m^2^ wheat field. Control soil samples (*n* = 3) were collected similarly as the rhizosphere samples from the control non-cultivated site 50 m away from the wheat field. Each replicate (*n* = 3) was a composite sample from four different locations within the site. Each soil sample comprising up to 15 g of soil was placed in an individual sterile plastic bag and transported to the laboratory in a cooler to avoid heating during the passage to the laboratory. All soil samples were sieved (mesh size: 2 mm) to remove root particles and other organic debris before biological analyses were undertaken.

### 2.2. Substrate Utilization

Before the substrate’s amendment, 10 g of soil samples were added to 15 sterile scintillation vials and acclimatized for 48 h at 26 °C, while being adjusted to moisture contents equivalent to 35% of their WHC (Water Holding Capacity). Soil sample from the same batch (replicate) was used for all the substrates (*n* = 3). An amount of 625 µL from each one of the substrates (Table 1) was added to each one of the soil sterile scintillation vials, yielding a total of 45 vials. The procedure followed a protocol similar to the SIR (soil-induce respiration) plates protocol. The soil with the substrates was stirred gently and left open for one hour to allow acidic substrates to eliminate CaCO_3_-derived CO_2_ evolution [22]. At the end of one hour, the vials were sealed, and after one hour, 500 mg was transferred individually into the Eppendorf tube and placed at −20 °C for DNA extraction.

### 2.3. DNA Extraction, Amplicon Sequencing, and Sequence Processing

Soil samples from the same batch of soil used for SIR were used further for microbial community composition (*n* = 3) analysis as follows. For scintillation, sterile vials of 2 g of soil were added to 125 µL of each one of the 15 substrates. The procedure was similar to the SIR plates’ protocol. At the end of one-hour incubation, soil samples were transferred individually into an Eppendorf tube and placed at −20 °C for DNA extraction.

DNA extraction: soil DNA was extracted from 0.5 g of soil sample from each one of the three samples (*n* = 2) using the GeneAll^®^ ExgeneTM mini-Kit (Seoul, Korea), according to the manufacturer’s instructions. The amplified DNA was stored at −20 °C until sequencing.

Bacterial taxonomic analysis—a pair of primers was used in order to express the bacteria region. PCR amplification was performed, using SimpliAmpTM thermal cycler (Thermo Fisher Scientific, Walham, MA, USA), by mixing 12.5 μL HS Taq Mix Red (PCR Biosystems, London, UK), 9.5 μL ultrapure water, 1.0 μL extracted DNA, 1 μL CS1-515F (ACACTGACGACATGGTTCTACAGTGCCAGCMGCCGCGGT), and 1 μL CS2-806R (TACGGTAGCAGAGACTTGGTCTGGACTACHVGGGTWTCT). The thermal cycling program was set to 95 °C for 3 min, 24 cycles of 98 °C for 10 s, 55 °C for 10 s, 72 °C for 20 s, and after the cycles, 72 °C for 1 min. The amplified DNA was stored at −20 °C until sequencing.

All final PCR products were run on an agarose gel to verify amplification specificity and quality, in parallel with a negative control. The final PCR products were conducted by Hylabs Inc. (Rehovot, Israel), using the Fluidigm Access Array primers for Illumina (San Diego, CA, USA), to generate libraries compatible for sequencing on the Miseq. Samples were measured for concentration by Qubit (ThermoFisher Scientific, Waltham, MA, USA) and size by Tapestation (Agilent Technologies, Santa Clara, CA, USA) and then sequenced on the Illumina Miseq using a Miseq V2 sequencing kit (500 cycles) to generate 2 × 250 paired end reads. The data were de-multiplexed using the Illumina base space cloud to generate two FASTQ files for each sample. The FASTQ files were imported into CLC-bio and analyzed as follows: Reads were trimmed for quality and adaptor sequences, merged, and then subjected to OTU picking to generate abundance tables. The database used for the OTU picking was Greengenes v13_5 at 97% sequence identity [23].

## 3. Data Analysis

The sequencing data were de-multiplexed using the Illumina base space cloud, to generate two FASTQ files for each sample. The FASTQ files were imported into the CLC bio–Genomics Workbench and analyzed with the Data QC, OTU clustering and Taxonomic Profiling workflows (Qiagen, CLC Bio, Aarthas, Denmark). Reads were trimmed for quality and adaptor sequences, merged, and then subjected to OTU clustering to generate abundance tables. The database used for the taxonomic assignment was Unite V7.2 at 97%.

### Statistical Analyses

All data (OTUs) were subjected to statistical analysis of variance using the SAS model (ANOVA and Duncan’s multiple range test) and were used for evaluating differences between separate means. A *p* < 0.05 value was considered significant [24].

In order to calculate the various alpha-diversity indices, we used the vegan (v2.6–4) package in R (v4.2.2). Using the vegan package in R, we calculated the Bray–Curtis distances between samples and plotted to PCoA (Principal coordinates analysis) using the ggplot2 package. Abiotic parameters were tested for association with community composition using the ‘envfit’ function in vegan.

The relative abundance of each microbial community composition was determined to the clustered OTU based on the number of OTUs.

We utilized PICRUSt2 v2.5.1 (phylogenetic investigation of communities by reconstruction of unobserved states) [25] to project the functional metagenome of the samples. The OTU table was converted to biom format using the ‘biom convert’ command to create a file compatible with the PICRUSt2 program. Initially, OTUs were phylogenetically placed using the ‘place_seqs.py’ script within PICRUSt2. Subsequently, gene family content was predicted using the hsp.py script. Finally, MetaCyc pathway abundance was projected using the pathway_pipeline.py script. The scripting functions within PICRUSt2 rely on the following tools: EPA-NG and gappa (place_seqs.py), Caster (hsp.py), and MinPath (pathway_pipeline.py).

Following the completion of the PICRUSt2 analysis, the resulting output files were then uploaded to the Statistical Analysis of Metagenomic Profiles (STAMP) v2.1.3 [26] software package. This software facilitates further statistical analyses of all predicted functional datasets and can be utilized to generate graphical representations of essential functional pathway data.

## 4. Results

### Soil Bacteria

The soil bacterial community composition related to the substrate at the phylum level (relative abundance > 1%) indicates which dominant bacterial groups are actively involved in substrate utilization.

A total of 12 phyla were obtained in the present study (Figure 1) where the six most dominant phyla were the *Proteobacteria* (30–51%) followed by *Bacterioidetes* (7–21%), *Acidobacteria* (8–15%), *Actinobacteria* (6–15%), *Chloroflexi* (7–11%), and *Gemmatimonadetes* (6–10%). The total number of orders obtained was 188, of which only 41 were known, and the remained 147 orders were unknown, including a total of 309 families, 485 genera, and 522 species, of which less than 10% were identified.

Of the total 12 phyla, seven phyla showed activity present in all the 16 substrates, where five phyla—*Planctomycetes* active in 15, *Cyanobacteria* active in 12, *Firmicutes* in 9, TM7 in 8, and *Nitrospirae phyla*—were triggered only by two substrates, one Oxalic and one Aromatic (Figure 1). Two phyla, the *Firmicutes* and *Nitrospirae,* were found not to respond to water amendment; moreover, the *Nitrospirae* were not found to use carbohydrates, which is known to be one of the most important energy sources.

Detecting the substrate usage of the 41 orders we found (Figure 2 Order), of which only fifteen orders were able to use all the 16 substrates; four orders used 15 substrates, where 14% orders used between 10 and 15 substrates, and the remaining 41% of the orders used between 1 and 10 substrates. Three orders, Herpetosiphonales, C114, and TM7-2 unknown order, were found to use only one substrate, e.g., Gamma amino butyric acid, 3,4 Dihidroxybenzoic acid, and L-Lysine, respectively.

Analyzing the profile of substrate utilization (Figure 3) can be used to estimate the intensity of competition and the overlap between the organisms. Orders with relatively higher capability of substrate use appeared to have a higher potential to compete for substrates rather than the Herpetosiphonales order, which is able to fulfill all its functionality only when a D-Fructose source is available.

In order to measure the variations in substrate utilization among the four groups, we analyzed each of the phylum groups using the Kruskal–Wallis test. This test helped us determine if there are statistically significant differences in substrate utilization between the four groups within each of the phylum taxonomical groups (Kruskal–Wallis test, see Table 2).

From the total of eleven phyla, only six were found to be affected by the four substrates, e.g., *Proteobacteria*, *Acidobacteria*, *Bacteroidetes*, *Actinobacteria*, and *Gematimonade* (Table 2). Each one of the species was found to yield descriptive statistics where FWER represents the family-wise error rate, which is the probability of making false discoveries, and q value provides a measure of each feature’s significance. Graphical representation of the Kruskal–Wallis test indicates that the distribution of the five substrates varies depending on the eleven phyla’s usage. The results of the Kruskal–Wallis tests are presented in Figure 4. The results of pairwise comparison confirmed that higher values obtained for one of the five substrates related to one phylum will be statistically significant. As a result of this, the *Acidobacteria* phyla were found to significantly (q = 0.00019) correlated with carboxylic acid substrate usage, and *Actinobacteria* significantly correlated with aromatic carboxylic acid substrate usage; *Bacteriodetes* correlated significantly (q = 0.00013) with amino acid, and *Actinobacteria* correlates significantly (q = 0.00019) with Aromatic Carboxylic acid. *Gemmatimonade* phyla was found to show a significant lower usage of carboxylic acid in comparison to the other three substrates. *Planctomycetes* phyla showed correlations q = 0.00761 with carboxylic acid that were lower in comparison to the other three substrates; for *Proteobacteria,* the trend was similar as before with q = 0.00015.

PICRUSt2 v2.5.1 was used to predict the functional potential of the soil microbiome under different agro-managements. A total of 229 significant pathways were predicted using PICRUSt2 analysis, and the predicted pathways were found to be significant using STAMP v2.1.3. A total of 442 predicted functional pathways were found belonging to 14 main categories (Figure 5). Each of the 14 categories is composed of a different number of functional profiles. Each profile category is associated with several important pathways that were shown to have significant differences in strength or intensity between different agro-management practices. No significant differences were obtained between each of the 14 categories, elucidating the importance of each one. The metabolic processes were represented by 61 categories, representing one of the most important processes in physiological functions and maintaining homeostasis in living organisms.

## 5. Discussion

The present study aims to investigate the changes in the soil microbial community in response to different substrates added to the soil. We hypothesized that the addition of different substrates will stimulate the growth of specific microbial species capable of consuming the respective substrate. The abundance of microbial taxa will be expressed by the specific substrate utilization of bacteria species that is expressed quantitatively in DNA. The data values obtained will allow us to determine the bacterial community composition and function related to each one of the substrates supplied. Each one of the 15 substrates belonging to one of the four groups changes accordingly [27]. Our results showed that bacterial relative abundance differs in response to the different substrates. Nitropirae was the only phylum that used only two substrate sources, carboxylic acid (20%) and aromatic acid (80%), there was relative abundance of Cyanobacteria, Firmicutes, and TM4 four substrates without the signs of usage of carboxylic acid, water-control and aromatic acid, respectively. There is a need to emphasize that the contribution of different phylum abundance being lower may mean that no sequences were detected. According to [28], the presence of the above phyla is because they were the most abundant phyla in soils. Upon examining the overlap in substrate utilization profiles among different microbial communities, we observed that higher substrate usage is indicative of a greater potential for competitive interactions. Our findings suggest a positive correlation between these interactions, as evidenced by the changes in relative abundance of various microbial phyla in our study [29,30]. Low-molecular-weight substrates, such as glucose, represent the most labile carbon (C) entering soil ecosystems [31]. Koch [32] in his review manuscript elucidated the adaptivity of some bacteria belonging to oligotrophs in a unique environment due to their specific adaptation. Many ‘copiotrophic’ bacterial species, however, are programmed to be able to adapt to several habitats. This means (and we know this applies to many other bacterial species) that these organisms are competent to adapt a wide range of life strategies for survival to fulfill life functions.

According to the copiotroph–oligotroph dichotomy frame for differentiating microbial life histories [33], copiotrophic bacterial phyla, such as *Proteobacteria*, *Bacteroidetes* and *Acitnobacteria* preferentially utilize glucose-C in a number of amendments following the initial addition of ^13^C-glucose, which was shown by their enrichment in heavy DNA fractions at certain sampling points [34,35,36,37,38]. In comparison, the oligotrophic bacteria phyla, such as *Acidobacteria* or *Chloroflexi*, were less enriched in the heavy DNA fractions, which indicated a lower ability to utilize labile C [16,17,37]. However, the changing internal competition and interaction among the bacterial taxa critically controls the activation. Therefore, the dynamic response of soil bacteria to resource input, rather than an instantaneous snapshot, would be more relevant to infer bacterial life strategies for substrate utilization and function in SOC (soil organic carbon) transformation [36,39,40,41]. However, the holistic imprint of exogenous substrates on the functional traits of bacteria was rarely considered.

Since soil serves as a substantial reservoir for chemical–physical and micronutrient components, it is subject to a wide array of environmental complexities and interactions enhancing multiple biotic components’ interaction simultaneously [42,43,44]. At the same time, the soil microbial community has numerous options for organizing phyla or species interaction such as decomposition, predation, commensalism or mutualism, and competition toward a complex model that will be able to effectively confront any challenging substrates.

It is crucial to gather field experimental data to provide empirical evidence, supported by a theoretical framework, to enhance our capacity to make predictions relevant to species coexistence and extinction, as well as to compare ecosystem processes when multiple species interact (Carrara et al., 2015 [43]). This has significant implications for ecosystem management, particularly in selecting species combinations that are not only more productive in the short term but also exhibit greater stability over long time frames.

## 6. Conclusions

We assessed how bacterial taxa responded to various substrate amendments, including aromatic acids, amino acids, carbohydrates, and carboxylic acids. Substrate-induced respiration (SIR) results showed that the functional traits of soil microorganisms were crucial for carbon (C) utilization. The ability to metabolize these substrates indicates a greater potential for competitive interactions within the microbial communities.

Shifts in bacterial diversity caused by substrate amendments, which impact the structure, dynamics, and functions of soil microbial communities, are vital for evaluating soil quality in sustainable agriculture. Understanding these changes is key to maintaining soil health and ensuring long-term agricultural productivity.

## Figures and Tables

**Figure 1 microorganisms-12-02034-f001:**
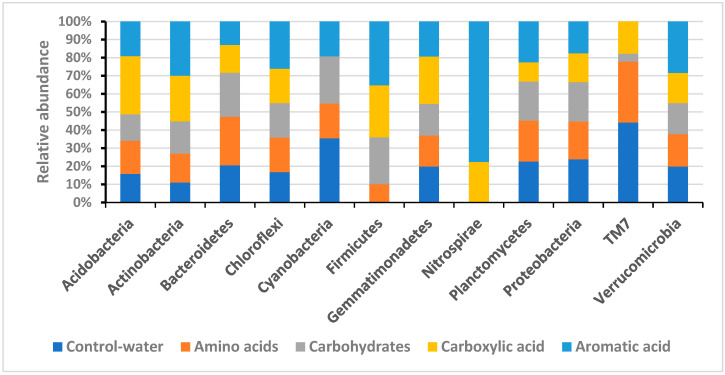
Relative abundance (%) of bacteria phyla in response to the four groups of substrate amendments.

**Figure 2 microorganisms-12-02034-f002:**
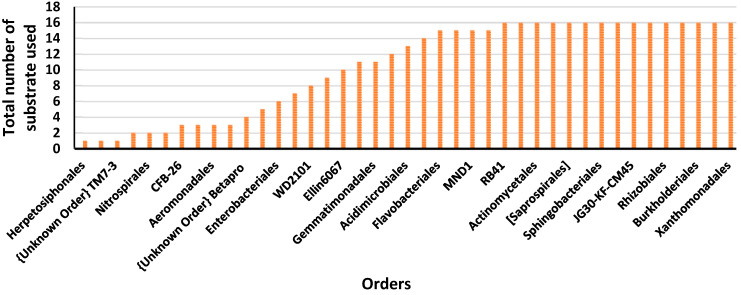
The total number of substrates used by each one of the forty-one orders.

**Figure 3 microorganisms-12-02034-f003:**
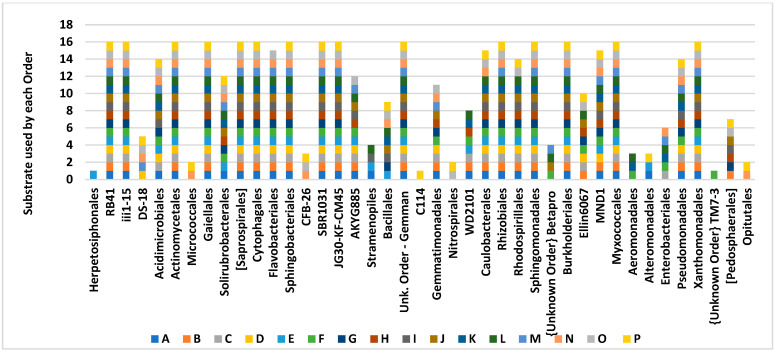
Elucidating the usage substrate by each order along the SIR study. A—Blank; B—L-Alanine; C—Arginine; D—L-Cysteine; E—Gamma amino butyric acid; F—L-Lysine; G—N-acetyl glucosamine; H—L-arabinose; I—D-Fructose; J—D-Glucose; K—Trehalose; L—D-Galactose; M—Citric acid; N—L-Malic acid; O—Oxalic acid; P—3,4 Dihidroxybenzoic acid.

**Figure 4 microorganisms-12-02034-f004:**
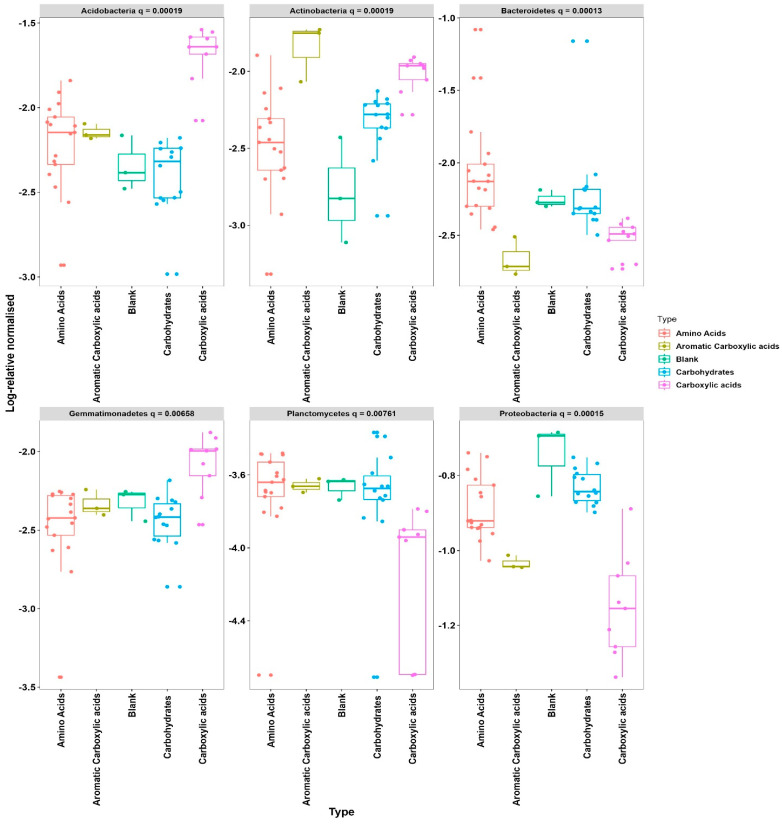
Graphical presentation of the four phyla related to the four substrate sources.

**Figure 5 microorganisms-12-02034-f005:**
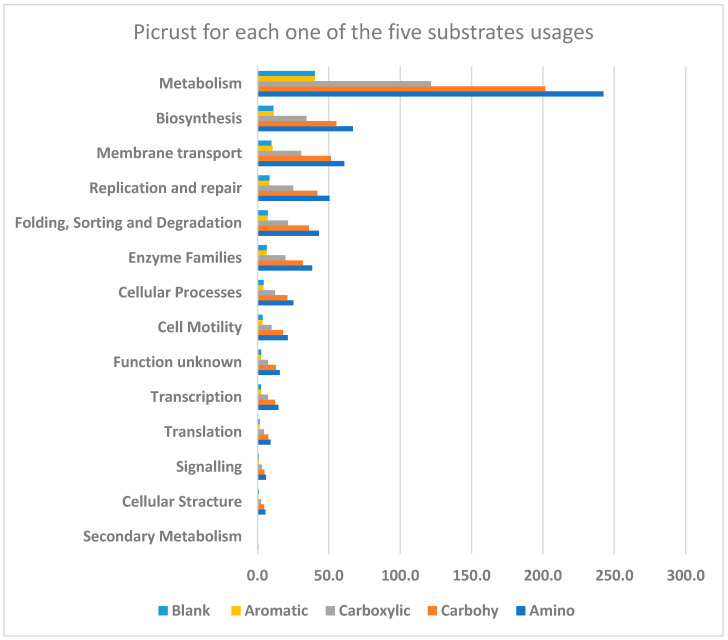
Functional categories in soil microbial communities, indicating the average proportion (%) of functional categories in soil microbial communities related to each of the five substrate usages.

**Table 1 microorganisms-12-02034-t001:** The fifteen different carbon sources including carbohydrates, carboxylic acids, amino acids and aromatic carboxylic acids used.

Carboxylic Acids	Carbohydrates	Amino Acids	Aromatic Acids
Citric acid	L-Arabinose	L-Alanine	3,4 Dihidroxybenzoic acid
L-Malic acid	D-Fructose	Arginine	
Oxalic acid	D-Glucose	L-Cysteine HCI	
	Trehalose	Amino Butiric acid	
	D-Galactose	L-Lysine	
		N-acetyl-Glucosamine	

**Table 2 microorganisms-12-02034-t002:** Kruskal–Wallis table for each one of the eleven phylum taxonomical levels for four substrate usage.

id	*p*	E Value	FWER	q Value Factor	q Value
Proteobacteria	1.36 × 10^−5^	0.00015	0.00015	11	0.00015
Acidobacteria	3.45 × 10^−5^	0.00038	0.000379	5.5	0.00019
Bacteroidetes	3.50 × 10^−5^	0.000385	0.000385	3.666667	0.000128
Actinobacteria	6.91 × 10^−5^	0.00076	0.00076	2.75	0.00019
Planctomycetes	3.46 × 10^−3^	0.038045	0.037394	2.2	0.007609
Gemmatimonadetes	3.59 × 10^−3^	0.039462	0.038762	1.833333	0.006577
Cyanobacteria	2.97 × 10^−2^	0.326927	0.28243	1.571429	0.046704
Chloroflexi	4.15 × 10^−2^	0.456824	0.372857	1.375	0.057103
Nitrospirae	6.44 × 10^−2^	0.708165	0.519049	1.222222	0.078685
Firmicutes	1.49 × 10^−1^	1.637842	0.830247	1.1	0.163784
TM7	2.27 × 10^−1^	2.498659	0.941245	1	0.227151

## Data Availability

The data presented in this study are openly available in [NCBI] [Read Archive database] [PRJNA924982]. The data presented in this study are available on request from the corresponding author.

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
