# Peer review of "Are Changes Occurring in Bacterial Taxa Community and Diversity with the Utilization of Different Substrates within SIR Measurements?"

_microorganisms, 2024, doi:10.3390/microorganisms12102034_

Round 1

Reviewer 1 Report

Comments and Suggestions for Authors

The subject of this article is interesting and brings new values ​​to the research on functional features of bacteria involved in soil organic carbon turnover. The authors proved that the measures of taxonomic composition and diversity of bacteria gradually changed in response to the supply of each SIR substrate. Moreover, the addition of organic matter promotes copiotrophic bacteria that increase soil fertility.

I have several comments on this manuscript:

Methodology. In this chapter, the reader is somewhat dissatisfied with the imprecise description of soil sampling from the study site. It should be so that every researcher from another country can conduct a similar comparative experiment (soil samples in different climatic and soil conditions). I do not learn from this methodology how many soil samples were actually taken? Was there any scheme regarding the sampling locations (so that their randomness was representative)?

In addition, the authors do not describe at least briefly the technique of sampling these samples (what device, cylinder, etc.)?

Conclusions? – the essence of the conclusions obtained from the research is not clearly articulated in this manuscript. We do have mentions of this in the “Abstract” chapter, as well as in the “Results” chapter.

However, there is no “Conclusions” which is key for me in scientific articles – in this chapter, one should refer to the research hypothesis adopted by the authors (has it been confirmed/refuted/partially confirmed? Then, one should provide the most important conclusion/conclusions from the conducted research. At the end of this chapter, one should provide recommendations/suggestions for further research in the scope of the subject of the article with a global (worldwide) impact.

After the authors have supplemented the changes (improvements) suggested by me, the article can be published in Microorganisms.

Author Response

Dear reviewers, 

Many thanks for all the comments and suggestions - which were of great help to improve the manuscript.  Most of both reviewer's comments have been taken into consideration - as you will find in the highlighted - red changes. 

I am away from my desk - and hope that all the changes that you requested will be accepted and the manuscript will be accepted for publication. 

Many thanks for your great work which I greatly appreciate. 

Reviewer 2 Report

Comments and Suggestions for Authors

The manuscript entitled “Are Changes Occurring in Bacterial Taxa Community and Diversity along with Utilization of Different Substrates—Within SIR Measurements?” is an interesting review about lactic acid and silage fermentation. Some details are recommended for the authors:

1.       Rephrase the abstract, as it does not reflect the work carried out but rather the reflections and assumptions made by the authors, which are given too much weight.

2.       In the introduction and discussion, provide greater support for the proposed work, as considering only the soil microbial community as copiotrophic/oligotrophic could be seen as a simplification of the event described.

3.       In materials and methods, section 2.1 requires more information on how and where the soil samples were taken, how many sites were sampled, what methodology was used, and specify the time at which the samples were taken. In section 2.2, explain the methodology more clearly and descriptively. The first paragraph of section 2.3 is repetitive with section 2.2. The phrase "a pair of primers was used in order to express the bacteria region" is repetitive, as the primers are described later. The data analysis procedure in section 2.3 is repetitive with the first paragraph of section 3. Specify which data were analyzed with an ANOVA.

4.       In the results, the first sentence of the section is incomplete.

5.       Include the definition of abbreviations like SIR, SOC, etc. Ensure that all the numbers of the molecules are written as subscripts.

6.       Check that all the references cited in the manuscript are in the reference section, and adjust the in-text citations according to the journal's guidelines.

7.       The entire document needs a thorough writing review.

Author Response

(The authors gave the same response as above.)

Round 2

Reviewer 2 Report

Comments and Suggestions for Authors

The manuscript entitled “Are Changes Occurring in Bacterial Taxa Community and Diversity along with Utilization of Different Substrates—Within SIR Measurements?” is an interesting review about lactic acid and silage fermentation. Some details are recommended for the authors:

 1.       Rephrase the abstract, as it does not reflect that it corresponds to a research work where the obtained results are reported.

2.       In the introduction provide greater support for the proposed work, as considering only the soil microbial community as copiotrophic/oligotrophic could be seen as a simplification of the described event.

3.       In materials and methods, in section 2.2, what is the "Soil Induce Respiration protocol"? In section 2.3, the phrase "To scintillation sterile vials 2 gr of soil were added to 125 μl of each one of the 15 substrates was added" is not clear. The phrase "a pair of primers was used in order to express the bacteria region" is repetitive, as the primers are described later.

4.       Check that all the references cited in the manuscript are in the reference section, and adjust the in-text citations according to the journal's guidelines. For example, Strickland et al., 2012 is not in the reference section.
